# Cracking the Endothelial Calcium (Ca^2+^) Code: A Matter of Timing and Spacing

**DOI:** 10.3390/ijms242316765

**Published:** 2023-11-26

**Authors:** Francesco Moccia, Valentina Brunetti, Teresa Soda, Roberto Berra-Romani, Giorgia Scarpellino

**Affiliations:** 1Laboratory of General Physiology, Department of Biology and Biotechnology “L. Spallanzani”, University of Pavia, 27100 Pavia, Italy; valentina.brunetti01@universitadipavia.it (V.B.); giorgia.scarpellino@unipv.it (G.S.); 2Department of Health Sciences, University of Magna Graecia, 88100 Catanzaro, Italy; teresa.soda@unicz.it; 3Department of Biomedicine, School of Medicine, Benemérita Universidad Autónoma de Puebla, Puebla 72410, Mexico; rberra001@hotmail.com

**Keywords:** endothelial cells, vasorelaxation, angiogenesis, Ca^2+^ signaling, Ca^2+^ sparklets, Ca^2+^ pulsars, Ca^2+^ wavelets, Ca^2+^ oscillations, Ca^2+^ waves, myo-endothelial gap junctions

## Abstract

A monolayer of endothelial cells lines the innermost surface of all blood vessels, thereby coming into close contact with every region of the body and perceiving signals deriving from both the bloodstream and parenchymal tissues. An increase in intracellular Ca^2+^ concentration ([Ca^2+^]_i_) is the main mechanism whereby vascular endothelial cells integrate the information conveyed by local and circulating cues. Herein, we describe the dynamics and spatial distribution of endothelial Ca^2+^ signals to understand how an array of spatially restricted (at both the subcellular and cellular levels) Ca^2+^ signals is exploited by the vascular intima to fulfill this complex task. We then illustrate how local endothelial Ca^2+^ signals affect the most appropriate vascular function and are integrated to transmit this information to more distant sites to maintain cardiovascular homeostasis. Vasorelaxation and sprouting angiogenesis were selected as an example of functions that are finely tuned by the variable spatio-temporal profile endothelial Ca^2+^ signals. We further highlighted how distinct Ca^2+^ signatures regulate the different phases of vasculogenesis, i.e., proliferation and migration, in circulating endothelial precursors.

## 1. Introduction

Once regarded as a mere physical barrier between the bloodstream and surrounding tissues, the vascular endothelium is now recognized as a distributed endocrine organ with a surface of 3000–7000 m^2^ and an estimated weight of 1 kg in humans [1,2,3]. Being located at the interface between flowing blood and parenchymal cells, vascular endothelial cells serve as a signal transduction platform that is simultaneously exposed to a myriad of chemical and physical inputs [4], which are integrated through an increase in intracellular Ca^2+^ concentration ([Ca^2+^]_i_) to generate the most appropriate output. Consequently, endothelial Ca^2+^ signals regulate a wide range of cardiovascular functions, including vascular resistance and permeability, coagulation, leukocyte transmigration, neurovascular coupling, angiogenesis and vasculogenesis [4,5,6,7,8,9,10,11]. Therefore, even a subtle dysregulation of endothelial Ca^2+^ signals can potentially lead to major human diseases [12,13,14,15,16,17,18,19,20,21,22,23,24], including hypertension, atherosclerosis, type 2 diabetes, obesity, deep vein thrombosis, sepsis, neurodegenerative disorders and cancer.

Vasodilation and sprouting angiogenesis represent the two most widely investigated functions regulated by endothelial [Ca^2+^]_i_ [4,5,8]. The mechanisms whereby endothelial Ca^2+^ signals regulate vasodilation, i.e., an increase in lumen diameter to increase blood flow to downstream vessels, include the gasotransmitter nitric oxide (NO) and endothelium-dependent hyperpolarization (EDH) via small/intermediate conductance Ca^2+^-activated K^+^ channels (K_Ca_), K_Ca_2.3 (SK_Ca_) and K_Ca_3.1 (IK_Ca_). However, the contribution of NO and EDH to endothelium-dependent vasodilation can vary along the vascular bed with NO being more effective in large conduit arteries and EDH in small resistance vessels [5,10]. Sprouting angiogenesis, i.e., the multistepped process that expands the existing vascular network through endothelial cell proliferation, migration and interaction with the surrounding environment, is also tuned by endothelial Ca^2+^ signals [8]. Vascular endothelial growth factor (VEGF), which is indispensable to initiate and direct the vascular sprout [8], may trigger a complex repertoire of Ca^2+^ signals to engage distinct Ca^2+^-sensitive pathways and regulate the different phases of the angiogenic response [8]. These examples embody the concept that endothelial Ca^2+^ signaling can recruit different Ca^2+^-dependent effectors to elicit specific vascular responses either depending on the vascular bed (e.g., vasorelaxation, which is primarily driven by NO in large vessels and EDH in small resistance vessels) or on the vascular function (e.g., vasorelaxation vs. angiogenesis) [4,25,26,27].

Herein, we provide an overview of how vascular endothelial cells regulate two representative examples of vascular functions, such as vasomotion and angiogenesis, by extracting the relevant information from the subcellular spatial profile and/or the temporal pattern of the underlying Ca^2+^ signal.

## 2. An Introduction to Endothelial Ca^2+^ Signaling: From Global to Local Ca^2+^ Signals

An increase in endothelial [Ca^2+^]_i_ can be elicited by neurohumoral agonists, e.g., acetylcholine [28,29,30,31] and adenosine trisphosphate (ATP) [32,33], inflammatory mediators, e.g., histamine [34,35] and bradykinin [36], growth factors, e.g., VEGF [37,38,39] and angiopoietins [40], and reactive oxygen species (ROS) [41,42]. Interestingly, endothelial Ca^2+^ waves can also be evoked by synaptically released glutamate [43,44] and γ-aminobutyric acid (GABA) [45,46] at the blood–brain barrier (BBB) as well as by sympathetic stimulation of the adrenergic terminals innervating the mesenteric bed [47,48,49,50,51]. In addition, endothelial Ca^2+^ signals can be elicited by physical stimuli, such as laminar shear stress [52,53,54], changes in intravascular pressure [55], mechanical scraping [56,57] and heat [58].

A kaleidoscope of intracellular Ca^2+^ waveforms can be elicited by chemical and neurohumoral stimulation of vascular endothelial cells, as extensively reviewed in [4,6,8,26,27,59,60]. The synthesis of Ca^2+^-sensitive fluorescent indicators, followed by the development of genetic Ca^2+^ indicators (GECIs), the advent of high-resolution microscopic imaging and the generation of endothelial-specific Cre mouse models, contributed to unveiling how endothelial cells assemble their versatile Ca^2+^ handling machinery to generate an array of spatiotemporally diverse Ca^2+^ signals that are selectively coupled to distinct effectors to activate precise vascular responses [25]. Initial epifluorescence Ca^2+^ imaging primarily conducted on vascular endothelial cells loaded with a ratiometric Ca^2+^ indicator (i.e., Fura-2) showed that the endothelial Ca^2+^ responses to physiological autacoids and growth factors consist of transient, biphasic, monotonic or oscillatory elevations in [Ca^2+^]_i_ that invade the whole cytoplasm and sometimes propagate to neighboring endothelial cells [25]. Nevertheless, high-speed Ca^2+^ imaging of endothelial cells loaded with non-ratiometric Ca^2+^ indicators (e.g., Fluo-3, Fluo-4 and Cal520) or expressing GECIs revealed that the physiological Ca^2+^ signals underlying vascular responses comprise a mosaic of spatially and temporally discrete Ca^2+^ events, which can remain highly localized to a specific subcellular site or propagate to generate intra- or intercellular endothelial Ca^2+^ waves [9]. These endothelial Ca^2+^ microdomains are mediated by the opening of Ca^2+^-permeable channels that are located both on the endoplasmic reticulum (ER) and on the plasma membrane (PM) and may be selectively coupled to distinct Ca^2+^-dependent effectors to modulate different functions.

### 2.1. General Mechanisms of Agonist-Induced Endothelial Ca^2+^ Signals

The endothelial Ca^2+^ response to chemical stimulation is primarily initiated by ER-dependent Ca^2+^ release and maintained by extracellular Ca^2+^ entry across the PM. ER Ca^2+^ mobilization requires agonist binding to G_q_-Protein Coupled Receptors (G_q_PCRs) or tyrosine kinase receptors (TKRs) on the PM, which are coupled to distinct isoforms of phospholipase C (i.e., respectively, PLCβ and PLCγ) [8,9,27,61]. PLC cleaves the PM-associated phosphoinositide 4,5-bisphosphate (PIP_2_) into diacylglycerol (DAG) and InsP_3_, which mobilizes ER Ca^2+^ by gating InsP_3_ receptors (InsP_3_Rs). InsP_3_Rs represent the major family of ER Ca^2+^-releasing channels in the endothelial lineage, as all the known InsP_3_R isoforms (i.e., InsP_3_R1, InsP_3_R2 and InsP_3_R3) are expressed in vascular endothelial cells [27,61,62]. PLC-derived InsP_3_ primes InsP_3_Rs to be activated by local Ca^2+^ such that, in the presence of permissive InsP_3_ concentrations, a spatially restricted Ca^2+^ pulse can be regeneratively propagated via Ca^2+^-induced Ca^2+^ release (CICR) [63]. The Ca^2+^-dependent recruitment of InsP_3_Rs could be supported by lysosomal Ca^2+^ release through two-pore channels (TPCs) [64,65], which may present two distinct isoforms (i.e., TPC1 and TPC2) in vascular endothelial cells and are gated by nicotinic acid adenine dinucleotide phosphate (NAADP). In contrast, ryanodine receptors (RyRs), which are mainly responsible for Ca^2+^ release from the sarcoplasmic reticulum (SR) in vascular smooth muscle cells (VSMCs) [27], are unlikely to significatively shape endothelial Ca^2+^ dynamics [8,9,27].

The InsP_3_-induced increase in endothelial [Ca^2+^]_i_ may rapidly return to the baseline due to the removal of cytosolic Ca^2+^ by Ca^2+^-clearing mechanisms, such as sarco-endoplasmic reticulum Ca^2+^-ATPase (SERCA), plasma membrane Ca^2+^-ATPase (PMCA), Na^+^/Ca^2+^ exchanger (NCX) and mitochondria. In most cases, however, the initial elevation in [Ca^2+^]_i_ may be followed by repetitive cycles of InsP_3_-induced ER Ca^2+^ release that occur as prolonged Ca^2+^ oscillations or by a sustained plateau level. In both cases, extracellular Ca^2+^ entry is required to refill the ER and prolong the Ca^2+^ response [4,6,8,25,26,27]. Agonist-induced Ca^2+^ influx in vascular endothelial cells can occur either through store-operated channels (SOCs) [61,66], which couple ER Ca^2+^ load with the Ca^2+^ permeability of the PM, or through members of the transient receptor potential (TRP) superfamily of non-selective cation channels that are activated downstream of the PLC pathway [67,68], such as TRP Canonical 3 (TRPC3) [69,70,71] and TRP Vanilloid 4 (TRPV4) [72,73,74]. In addition, agonist-induced Ca^2+^ entry in endothelial cells could be supported by ionotropic receptors, such as P2X receptors [32] and N-methyl-D-aspartate receptors (NMDARs) [44].

### 2.2. Endothelial Ca^2+^ Microdomains Generated by InsP_3_-Dependent ER Ca^2+^ Release

InsP_3_Rs are non-selective cation channels that present large single-channel conductance but only moderate Ca^2+^ selectivity over Na^+^ and K^+^ (P_Ca_/P_K_ ~7) [75]. The three InsP_3_R isoforms differ in their sensitivity to both InsP_3_- and Ca^2+^-dependent activation and, therefore, may encode distinct InsP_3_-mediated Ca^2+^ signals [76,77]: InsP_3_R2, which is the most sensitive to InsP_3_, sustains long-lasting oscillations in [Ca^2+^]_i_; InsP_3_R1 supports irregular Ca^2+^ oscillations; and InsP_3_R3, which is the less sensitive isoform to both InsP_3_ and Ca^2+^, serves as an anti-oscillatory unit that generates monophasic Ca^2+^ signals. To illustrate how this concept also applies to the vascular endothelium, acetylcholine and glutamate trigger intracellular Ca^2+^ oscillations in mouse cerebrovascular endothelial cells expressing InsP_3_R1 and InsP_3_R2 [43,78], while they induce a biphasic elevation in [Ca^2+^]_i_ in their human counterparts expressing only InsP_3_R3 [31,79]. The global Ca^2+^ waves elicited by endothelial autacoids result from the summation of multiple local Ca^2+^ events that differ in spatial distribution and duration and are known as blips, puffs, pulsars and wavelets (Figure 1). Because of their spatial distribution, some of these discrete Ca^2+^ events (i.e., pulsars and wavelets) can be detected in native, but not cultured, endothelial cells. Local Ca^2+^ events sustained by rhythmic InsP_3_R activation may also spontaneously occur in the vascular intima. However, they are more likely to take place in resistance-sized vessels as compared to large conduit vessels [28,29,80,81].

#### 2.2.1. Ca^2+^ Blips and Puffs in Cultured Endothelial Cells

Ca^2+^ puffs, which can also occur spontaneously [82,83], are the building blocks of the global Ca^2+^ signals elicited by physiological agonists (Figure 1). A preliminary characterization of the spatio-temporal organization of agonist-evoked changes in endothelial [Ca^2+^]_i_ was conducted on calf pulmonary artery endothelial (CPAE) cells (Table 1). Confocal scanning microscopy revealed that the most elementary endothelial Ca^2+^ release event is represented by the Ca^2+^ blip, which reflects the opening of a single InsP_3_R (Figure 1) [84]. Endothelial Ca^2+^ blips present an average amplitude of 23 nM and a spatial spread of 1–3 µm and last <100 ms (Table 1) [84]. Superposition of discrete Ca^2+^ blips results in agonist-evoked Ca^2+^ puffs, which derive from the opening of a fixed cluster of adjacent InsP_3_Rs and present a higher amplitude (50–100 nM vs. 23 nM) and wider spatial spread (30 µm vs. 1–3 µm) than unitary blips (Table 1) [84]. Endothelial Ca^2+^ puffs precede by several hundred milliseconds the regenerative Ca^2+^ wave that propagates from the periphery to the central region of the cell at an average speed of 20–60 μm/s upon supra-physiological chemical stimulation [84]. Subsequent reports have shown that agonist-evoked cytosolic Ca^2+^ waves can propagate into the nucleus through CICR or by diffusion through nuclear pores in several types of cultured vascular endothelial cells [85,86,87].

#### 2.2.2. Ca^2+^ Puffs, Pulsars and Wavelets in Native Endothelial Cells

Direct imaging of endothelial Ca^2+^ activity has confirmed that, in native macro- and microvascular endothelial cells, the regenerative Ca^2+^ wave elicited by physiological autacoids, such as acetylcholine [82,88,89,90] and adenosine [91,92], is initiated by the spatio-temporal recruitment of local Ca^2+^ puffs. InsP_3_-driven endothelial Ca^2+^, puffs in situ were first reported in rat ureteral arterioles [82] and then characterized in mouse small mesenteric precapillary arterioles [83] and rat tail artery [89], in which low concentrations of acetylcholine induced localized Ca^2+^ release events mainly originating at the distal ends of the cells. The spatio-temporal summation of InsP_3_-driven ER Ca^2+^ puffs leads to a global elevation of endothelial [Ca^2+^]_i_ that could spread to neighboring endothelial cells to ultimately coordinate the vascular function, as outlined in Section 2.4.1.

However, the opening of a spatially restricted cluster of Ca^2+^-permeable channels may also generate local Ca^2+^ microdomains that are located around the inner mouth of the open channel, do not generate cell-wide Ca^2+^ waves and are tightly coupled to specific Ca^2+^-dependent effectors. These local Ca^2+^ signals are known as Ca^2+^ pulsars, Ca^2+^ wavelets and Ca^2+^ sparklets (Figure 1 and Table 1). Endothelial Ca^2+^ pulsars and Ca^2+^ wavelets are subcellular Ca^2+^ signals that are detectable only in native cells because they occur specifically at myo-endothelial projections (MEPs) (Figure 1) [10], whereas Ca^2+^ sparklets occur at the PM (Figure 1) and will be described in Section 2.3. Endothelial cells may send projections, known as MEPs, through holes in the internal elastic lamina (IEL) that separates the intimal and medial layers of the vascular wall (Figure 1) [5,10,80]. MEPs enable a single endothelial cell to make direct contact with multiple VSMCs; furthermore, MEPs may couple adjoining endothelial cells and VSMCs via myo-endothelial gap junctions (MEGJs), which allow the transmission of both electrical and chemical (e.g., Ca^2+^ and InsP_3_) signals [5,10,80]. The abundance of MEPs is inversely related to the vessel diameter such that they are more abundant in small resistance arteries and arterioles as compared to large vessels [5,93]. Studies carried out in mouse mesenteric third-order arterioles have shown that MEPs are enriched with the pro-oscillatory Ca^2+^ release units, InsP_3_R1 and InsP_3_R2 [94], which generate spontaneous, spatially fixed Ca^2+^ signals, known as Ca^2+^ pulsars [80]. These Ca^2+^ pulsars display a wider spatial spread as compared to Ca^2+^ puffs (~14 µm^2^) and occur repeatedly at the same sites with a frequency of 0.08 Hz (Table 1) [80]. Endothelial Ca^2+^ pulsars are driven by tonic PLC activity [95], but their frequency can be enhanced by the stimulation of G_q_PCRs, including muscarinic M3 receptors [80] and protease-activated receptor-2 (PAR-2) [96]. The mechanisms responsible for tonic PLC activation in pressurized resistance arteries are still unclear, but the spontaneous local Ca^2+^ activity recorded in the presence of physiological shear stress was due to flow-induced acetylcholine release and M3 receptor activation [29]. Nevertheless, an excessive increase in intraluminal pressure may suppress InsP_3_-induced Ca^2+^ release events by altering the endothelial cell shape and geometry, thereby limiting cytosolic Ca^2+^ diffusion that drives CICR through InsP_3_Rs [97]. The inhibitory effect of intraluminal pressure could decrease with aging, thereby leading to an increase in InsP_3_-induced ER Ca^2+^ release events [98]. Recruitment of Ca^2+^-calmodulin (CaM)-dependent protein kinase II (CaMKII) by the oscillatory Ca^2+^ signals may dampen the frequency of Ca^2+^ pulsars by inhibiting InsP_3_Rs, thereby acting as an additional feedback inhibition mechanism to fine-tune the spontaneous endothelial Ca^2+^ activity [94].

New endothelial Ca^2+^ pulsar sites can also be activated by sympathetic nerve stimulation of adjoining VSMCs via Ca^2+^ or InsP_3_ transfer through MEGJs in pressurized resistance arteries or arterioles [48,49]. The engagement of VSMC α1-adrenergic receptors may lead to InsP_3_ production within VSMCs overlying the endothelial monolayer; the newly formed InsP_3_ may then diffuse through MEGJs, thereby increasing the frequency of spontaneous Ca^2+^ pulsars and activating endothelium-dependent vasorelaxing pathways (i.e., EDH and NO) [49]. This mechanism was termed myo-endothelial feedback and could be mimicked by the pharmacological stimulation of arteriolar VSMCs with phenylephrine [99]. Subsequently, it was shown that VSMC depolarization was per se able to accelerate the frequency and increase the spatial distribution of endothelial Ca^2+^ pulsars in a voltage-dependent manner [48,81]: according to this model, the opening of voltage-gated L-type Ca^2+^ channels in arteriolar VSMCs leads to extracellular Ca^2+^ entry, which passes through MEGJs to stimulate endothelial InsP_3_Rs via CICR [48]. The endothelial Ca^2+^ signals that occur during myo-endothelial feedback are regulated by non-ER-based calreticulin, although the underlying mechanism is still unknown [100]. The myo-endothelial feedback could also be mediated by another type of local endothelial Ca^2+^ signals, known as Ca^2+^ wavelets, in hamster skeletal muscle arteries [50]. Endothelial Ca^2+^ wavelets can be activated at MEPs by rhythmic ER Ca^2+^ release through InsP_3_Rs upon stimulation of VSMC α1-adrenergic receptors with phenylephrine [50]. With respect to Ca^2+^ pulsars (Table 1), Ca^2+^ wavelets present a longer duration (~0.47 s vs. ~270 ms), a larger area (~41 µm^2^ vs. ~14 µm^2^) and a greater frequency (0.22 Hz vs. 0.08 Hz) [50].

**Table 1 ijms-24-16765-t001:** Local Ca^2+^ signals in vascular endothelial cells.

Ca^2+^ Signal	Source	Spatio-Temporal Features and Amplitude	Function	Reference
Ca^2+^ blips	InsP_3_Rs in the ER	Spatial spread: 1–3 µm; duration < 100 ms; amplitude: 23 nM	Building block of Ca^2+^ puffs	[84]
Ca^2+^ puffs	InsP_3_Rs in the ER	Spatial spread: 30 µm; duration: >>100 ms; amplitude: 50–100 nM	Building blocks of the intracellular Ca^2+^ waves	[84]
Ca^2+^ pulsars	InsP_3_Rs in the ER	Area: ~14 µm^2^; duration: ~250 ms; frequency: ~0.08 Hz	Recruitment of SK_Ca_/IK_Ca_ channels at MEPs and, to be demonstrated, of eNOS to induce endothelium-dependent vasorelaxation	[48,49,80,94]
Ca^2+^ wavelets	InsP_3_Rs in the ER	Area: ~41 µm^2^; duration: ~470 ms; frequency: ~0.22 Hz	Recruitment of SK_Ca_/IK_Ca_ channels at MEPs and, to be demonstrated, of eNOS to induce endothelium-dependent vasorelaxation	[50]
TRPV4-mediated Ca^2+^ sparklets	TRPV4 in the PM	Area: ~11 µm^2^; τ: ~37 ms; frequency: ~0.25 Hz	Recruitment of SK_Ca_/IK_Ca_ channels at MEPs to induce endothelium-dependent vasodilation in systemic resistance arteries; recruitment of eNOS to induce vasodilation in pulmonary resistance vessels	[47,53,55,101,102,103]
TRPV3-mediated Ca^2+^ sparklets	TRPV3 in the PM	Area: ~1–2 µm^2^; duration: ~70 ms; frequency: ~0.6 Hz	SK_Ca_/IK_Ca_ channels to induce endothelium-dependent vasodilation in cerebral parenchymal arterioles	[104,105]
TRPA1-mediated Ca^2+^ sparklets	TRPA1 in the PM	Area < 1 µm^2^; duration > 200 ms; frequency: ~0.3 Hz	SK_Ca_/IK_Ca_ channels and TRPA1/Panx1/purinergic signaling to induce endothelium-dependent vasodilation in cerebral parenchymal arterioles	[106,107,108,109]
NMDAR-mediated Ca^2+^ sparklets	NMDARs in the PM	Area: 8 µm^2^; duration: 300 ms and 500 ms; frequency: ~0.15 Hz	SK_Ca_/IK_Ca_ channels and eNOS to induce endothelium-dependent vasodilation in cerebral parenchymal arteries	[14,110]

Abbreviations: eNOS: endothelial nitric oxide synthase; MEPs: myo-endothelial projections; PM: plasma membrane; τ: time constant.

Finally, an array of ER-dependent local Ca^2+^ events was also recorded in the swine coronary artery (SCA) endothelium [95], in which the basal production of InsP_3_ induces spatially restricted Ca^2+^ signals (~34 µm^2^) that show a lower frequency (0.016 Hz) and a greater propensity to develop into cell-wide Ca^2+^ waves as compared to Ca^2+^ pulsars described in mouse mesenteric arteries [80,95]. Furthermore, local Ca^2+^ signals in the SCA endothelium did not arise in proximity to MEPs but around the nucleus, which presented the highest density of InsP_3_Rs [95]. As observed for Ca^2+^ pulsars, however, the frequency of these local Ca^2+^ events may be increased by stimulating SCA endothelial cells with substance P [95]. Unlike pulsars, the local Ca^2+^ events arising in the SCA endothelium were more likely to develop into regenerative Ca^2+^ waves due to the orientation of InsP_3_Rs along the longitudinal axis of the cells that favors directional CICR [95]. Thus, the spatial location and distribution of the endothelial InsP_3_-driven local Ca^2+^ events may differ along the vascular tree (e.g., pulsars in mesenteric arteries vs. Ca^2+^ wavelets in hamster skeletal muscle arteries or the wider Ca^2+^ signals occurring in the SCA).

A compelling issue that warrants further investigation is the contribution of lysosomal Ca^2+^ release through TPCs to InsP_3_Rs-mediated local Ca^2+^ signals. A recent series of studies suggested that NAADP-induced lysosomal Ca^2+^ release through TPCs is required to prime InsP_3_Rs for InsP_3_-dependent activation, thereby triggering ER Ca^2+^ release in the endothelial lineage [31,43,111]. However, ultrastructural analysis failed to detect lysosomal vesicles at the MEPs [112,113]. This evidence suggests that the basal Ca^2+^ levels in vascular endothelial cells may be higher in the proximity of MEGJs such that locally synthesized InsP_3_ does not require NAADP-induced TPC activation to trigger Ca^2+^ release from the ER.

### 2.3. Endothelial Ca^2+^ Microdomains Generated by Extracellular Ca^2+^ Entry

Extracellular Ca^2+^ entry generates local Ca^2+^ microdomains that are spatially restricted in close proximity to the inner mouth of the protein channel, known as Ca^2+^ sparklets (Table 1) [6,9], and were originally described in rat ventricular myocytes [114]. Endothelial Ca^2+^ sparklets are mediated by TRPV4 [101], TRPV3 [105] or TRP Ankyrin 1 (TRPA1) [107] and by NMDARs in cerebrovascular endothelial cells [14]. Local endothelial Ca^2+^ signals can also be generated by SOCs [61].

#### 2.3.1. Ca^2+^ Sparklets in Native Endothelial Cells: TRPV3, TRPV4 and TRPA1

Endothelial Ca^2+^ sparklets were first described by the Nelson group and found to be mediated by TRPV4 channels (Figure 1) [101]. TRPV4 is a polymodal non-selective cation channel that can integrate an array of chemical and physical cues [60,74]. These include, but are not limited to, phospholipase A2 (PLA2)-dependent generation of arachidonic acid (AA) and epoxyeicosatrienoic (EET) acids, PLC-dependent depletion of PIP_2_ or formation of InsP_3_ and DAG, shear stress, mechanical stretch and decrease in intravascular pressure [60,73,74,102,115,116]. Using high-resolution confocal microscopy to image local Ca^2+^ activity of an en face preparation of third-order mesenteric arteries explanted from a GCaMP2-expressing mouse, Sonkusare et al. recorded extracellular Ca^2+^ influx through single TRPV4 channels stimulated with the synthetic agonists GSK 1016790A (GSK) and 4α-Phorbol 12,13-didecanoate (4α-PDD) [101]. Additionally, submembrane Ca^2+^ microdomains were elicited by 11,12-EET, an endogenous ligand of TRPV4 channels. The endogenous TRPV4-mediated Ca^2+^ sparklets were not spike-like Ca^2+^ signals but presented a clear plateau phase, had a spatial distribution of ~11 µm^2^ and occurred repeatedly within the same subcellular sites, the majority of which were located within MEPs (Table 1) [101]. Quantal events of Ca^2+^ influx could be recorded, and, accordingly, the amplitude of TRPV4-mediated Ca^2+^ sparklets was sensitive to changes in the electrochemical gradient across the PM [101]. TRPV4 channels assembled into four-channel metastructures that were maintained by the kinase- and phosphatase-anchoring protein AKAP150 (A-kinase anchoring protein 150) (Figure 1), conferring them a cooperative gating behavior: local Ca^2+^ entry through an individual TRPV4 channel could be potentiated by the Ca^2+^-dependent activation of nearby channels [6,101,102]. Physiologically, TRPV4-mediated Ca^2+^ sparklets were triggered by acetylcholine via protein kinase C (PKC)-dependent phosphorylation, which required PKC anchoring to AKAP150 (Figure 1) [102]. The endothelial TRPV4-mediated Ca^2+^ sparklets could also be activated either by a drop in intravascular pressure below 50 mmHg [55] or by an increase in fluid shear stress below 4–8 dyne/cm^2^ [53]. In addition, TRPV4-mediated Ca^2+^ sparklets at MEPs could be indirectly activated upon stimulation of VSMC α1-adrenergic receptors with phenylephrine or thromboxane A2 receptors with U46619 [47]. Subsequently, InsP_3_ diffuses through MEGJs to the underlying endothelial cells, thereby activating TRPV4 in a Ca^2+^-independent manner [47].

Endothelial Ca^2+^ sparklets mediated by TRPV3 (Figure 1) and TRPA1 (Figure 1) were recorded in the rodent brain vasculature (Table 1). Ca^2+^ sparklets resulting from the opening of a single TRPV3 channel were activated by carvacrol, a typical ingredient of the Mediterranean diet (oregano), in rat pial and parenchymal endothelial cells [105]. However, this study did not evaluate whether TRPV3 channels are located at MEPs. Conversely, the endothelial TRPA1 was found to colocalize with NADPH oxidase 2 (NOX2) in the proximity of MEGJs in rat cerebral parenchymal arterioles (Figure 1) [107]. TRPA1-mediated Ca^2+^ sparklets occurred under basal conditions in pressurized arterioles, but their frequency was significantly increased by NOX2-generated reactive oxygens species (ROS) (Table 1), such as 4-hydroxy-nonenal (4-HNE) [107], a product of lipid peroxidation that can selectively gate TRPA1 [117,118], and Allyl Isothiocianate (AITC) [107], a pungent compound that is very abundant in cruciferous vegetables. The most common submembrane Ca^2+^ event was generated by the simultaneous opening of two TRPA1 channels [107]. TRPA1 channels have also been detected in human cerebrovascular endothelial cells, where they are gated by 4-HNE but not by AITC. However, TRPA1-mediated Ca^2+^ sparklets were not investigated in this study [119].

#### 2.3.2. Ca^2+^ Sparklets in Native Cerebrovascular Endothelial Cells: NMDARs

Native and cultured cerebrovascular endothelial cells express ionotropic receptors classically belonging to the neuronal lineage, such as NMDARs [44,110,120] and GABA sub-type A receptors (GABA_A_Rs) [45,46]. Interestingly, both NMDARs and GABA_A_Rs contribute to shaping the endothelial Ca^2+^ response to their physiological agonists, the excitatory neurotransmitter glutamate [44,120] and the inhibitory neurotransmitter GABA [45,46], respectively. The Pires group has characterized the unitary Ca^2+^ entry events through NMDARs in mouse cerebral artery endothelial cells, showing that they consist of submembrane Ca^2+^ sparklets with a spatial spread of 8 µm^2^ (Table 1) [14]. NMDAR-mediated Ca^2+^ entry in the human cerebrovascular endothelial cell line, hCMEC/D3, is not detectable by epifluorescence imaging of Fura-2 loaded cells but leads to robust NO release [120]. This observation led the authors to suggest that NMDARs also mediate local Ca^2+^ sparklets in hCMEC/D3 cells.

#### 2.3.3. Local Ca^2+^ Signals Activated by Store-Operated Ca^2+^ Entry (SOCE) in Vascular Endothelial Cells

Little information is available regarding the local Ca^2+^ signals induced by SOCE activation in the endothelial lineage [61]. SOCE is regarded as the major Ca^2+^ entry pathway that replenishes the ER Ca^2+^ store and sustains long-lasting oscillations in cultured endothelial cells [61,66] and in native macrovascular endothelial cells [20,121,122,123]. In accord, the distance between the ER and the PM in microvascular endothelial cells is significantly longer than in macrovascular endothelial cells, and the SOCE machinery is yet to be reported at MEPs [5]. Endothelial SOCE is activated upon InsP_3_-dependent reduction in ER Ca^2+^ concentration ([Ca^2+^]_ER_) and is mediated by the physical interaction between STIM1 (Figure 1), which serves as the sensor of [Ca^2+^]_ER_, and the Ca^2+^-permeable Orai1 channels (Figure 1) or the non-selective TRPC1/TRPC4 channels (Figure 1), which are located on the PM [61]. The store-operated current mediated by STIM1 and Orai1 channels has been termed Ca^2+^-release-activated Ca^2+^ current (I_CRAC_), whereas STIM1 and TRPC1/TRPC4 channels mediate a current known as I_SOC_. In addition, a heteromeric complex comprising STIM1, Orai1 and TRPC1/TRPC4 has been described in some endothelial cells. This complex mediates an I_CRAC_-like current with intermediate biophysical features between I_CRAC_ and I_SOC_ [61].

Local Ca^2+^ signals mediated by STIM1 activation at the leading edge of migration were recorded in cultured human umbilical vein endothelial cells (HUVECs) [124]. Furthermore, studies on CPAE cells have revealed that SOCE colocalizes with InsP_3_-induced ER Ca^2+^ release at sites of focal stimulation with sub-maximal doses of an agonist. However, SOCE can be activated at more remote locations by an increase in the strength of local stimulation because of intraluminal redistribution of ER Ca^2+^ [66,125]. Unfortunately, the measurement of local store-operated Ca^2+^ signals in native endothelial cells is still missing.

### 2.4. Intra- and Intercellular Endothelial Ca^2+^ Waves

The spatially restricted endothelial Ca^2+^ events that are generated by local Ca^2+^ puffs can be amplified by the recruitment of adjacent InsP_3_Rs via CICR and coalesce into a global elevation in [Ca^2+^]_i_, which can spread to neighboring endothelial cells, thereby originating both intra- (Table 2) and intercellular Ca^2+^ waves [4,9,10]. Extracellular Ca^2+^ entry through SOCs or TRP channels may sustain these regenerative fluctuations in endothelial [Ca^2+^]_i_, although the contribution of Ca^2+^ influx may be variable depending on the agonist or vascular bed (Table 2).

#### 2.4.1. Intracellular Ca^2+^ Waves in Vascular Endothelial Cells

Ca^2+^ imaging analysis of either pressurized or en face vascular preparations by using high-speed confocal microscopy has revealed that physiological concentrations of neurohumoral mediators, such as acetylcholine, histamine or ATP, induce intracellular Ca^2+^ oscillations in native endothelial cells to regulate blood pressure or microvascular permeability. Growth factors, such as VEGF, have also been shown to elicit endothelial Ca^2+^ oscillations to stimulate sprouting angiogenesis in vivo [8] Table 2 lists some of the most representative examples of agonist-induced intracellular Ca^2+^ waves in vascular endothelial cells.

Further work is required to assess the mechanisms supporting endothelial Ca^2+^ oscillations, which could vary depending on the vascular bed and the species (Table 1). Do lysosomal TPCs support ER Ca^2+^ release through InsP_3_Rs (see also Section 2.2.2 and Section 6)? Does an SKF-sensitive SOCE sustain endothelial Ca^2+^ oscillations in large conduit arteries (Section 2.3.3 and Section 4.1), while TRPV4 channels fulfill this role in systemic resistance vessels (see Section 2.3.1 and Section 4.2.1)? May the Ca^2+^ entry pathway vary with the agonist?

#### 2.4.2. Intracellular Ca^2+^ Waves Induced by Neuronal Activity in Cerebrovascular Endothelial Cells

A recent investigation revealed that synaptic activity can also trigger endothelial Ca^2+^ waves in the mouse brain microvasculature [131]. In vivo imaging in anesthetized transgenic mice selectively expressing GCaMP8 in endothelial cells revealed a hierarchy of InsP_3_-dependent Ca^2+^ signals, ranging from small, subsecond protoevents, caused by Ca^2+^ mobilization through a limited number of InsP_3_Rs, to high-magnitude, persistent (up to ~1 min) composite Ca^2+^ events sustained by large clusters of InsP_3_Rs [131]. These frequent InsP_3_-dependent local Ca^2+^ release events were sustained by TRPV4 activation upon PLCβ-dependent hydrolysis of PIP_2_ [131,132]. Somatosensory stimulation induced an increase in the amplitude and duration of these subcellular Ca^2+^ events and increased the prevalence of endothelial Ca^2+^ activity at the arteriole–capillary transitional zone [131]. The agonist responsible for G_q_PCR activation has not been identified [131,133], but it could be the synaptically released excitatory neurotransmitter glutamate [134]. In accord, glutamate can bind to metabotropic glutamate receptor 1 (mGluR1) and mGluR5 to induce intracellular Ca^2+^ oscillations in cultured cerebrovascular endothelial cells [43,79].

#### 2.4.3. Intercellular Ca^2+^ Waves in Vascular Endothelial Cells

Intracellular Ca^2+^ waves that occur either spontaneously [91,127] or upon physiological stimulation can propagate along the vascular intima of resistance vessels as intercellular Ca^2+^ waves [27,135]. Local application of up to 1 µM acetylcholine induced intercellular Ca^2+^ waves that spread to adjacent endothelial cells in resistance vessels, such as mouse abdominal arterioles [90], the mouse cremaster muscle artery [136,137] and the rat mesenteric artery [29]. PAR-2 stimulation could also trigger endothelial Ca^2+^ waves that propagate across adjacent cells in the rat mesenteric artery [138]. Using transgenic mice selectively expressing GCaMP2 in vascular endothelial cells, Tallini et al. found that the local application of 1 µM acetylcholine induced intercellular Ca^2+^ waves that propagated for ~1 mm along the vascular intima at a speed of ~116 µm/s [137]. However, when acetylcholine was administered within the same concentration range (<3 µM), the intracellular Ca^2+^ waves did not spread beyond a limited cluster of endothelial cells, as observed in large conduit arteries [28].

The propagation of intercellular Ca^2+^ waves in the vascular intima can be supported either by the diffusion of cytosolic second messengers [128,139], e.g., Ca^2+^ and/or InsP_3_, through gap junctions or by the paracrine release of extracellular mediators [140,141], e.g., ATP. An alternative mechanism has been proposed for the hemodynamic response to prolonged (5 s) somatosensory stimulation in the whisker barrel cortex. Neuronal activity has been found to trigger an endothelial Ca^2+^ wave in brain capillaries that travels back to upstream feeding arterioles, thereby resulting in functional hyperemia [109]. It has been suggested that active neurons release ROS that can peroxidize membrane lipids and thereby generate metabolites, e.g., 4-HNE, in the proximity of brain capillaries [109,134]. ROS-derived metabolites, in turn, gate TRPA1 and induce the Ca^2+^-dependent opening of adjacent pannexin 1 (Panx1) channels, which release ATP into the extracellular milieu. The local elevation in extracellular ATP directly activates the Ca^2+^-permeable P_2_X receptors on neighboring endothelial cells, thus triggering a slow Ca^2+^ wave that spreads from the distant capillaries to the post-arteriole transitional segment [109]. A Panx1/purinergic signaling pathway also controls capillary-to-arteriolar communication in the hamster skeletal musculature [142], but it is still unclear whether it is triggered by TRPA1-mediated Ca^2+^ sparklets. Therefore, exploring the mechanisms driving the propagation of intracellular Ca^2+^ waves is mandatory to fully appreciate how they control vascular function (see below).

## 3. Vascular Endothelial Cells Use Ca^2+^ Signals to Interpret the Local Microenvironment and Transfer Information to Distant Sites

Vascular endothelial cells are immersed in a stream of information that must be independently processed to generate the most appropriate vascular output. These cues consist of modest changes in the chemical and physical composition of the local microenvironment that elicit an elevation in [Ca^2+^]_i_ superimposed on the spontaneous Ca^2+^ activity, e.g., spontaneous Ca^2+^ puffs, also termed Ca^2+^ noise [4,26]. Therefore, endothelial cells need to reject the meaningless Ca^2+^ noise and extract the information that is encoded in the variety of Ca^2+^ signals induced by extracellular cues [4,26]. Recent studies have shown that vascular endothelial cells that are located at different positions of the vascular intima can detect distinct chemical signals due to the heterogeneity in the expression of their surface receptors. In addition, they have the potential to transmit local Ca^2+^ signals to more remote sites to coordinate specific vascular functions, such as vasodilation and local blood perfusion [4,26].

Immunofluorescence showed that purinergic P2Y2 receptors and muscarinic M3 receptors are segregated in spatially distinct clusters of endothelial cells in the rat carotid artery [143]. Functional analysis of Ca^2+^ activity in large populations of endothelial cells within native vessels confirmed that the endothelial Ca^2+^ response to extracellular stimuli is not homogeneous [4,26]. Acetylcholine evoked large Ca^2+^ signals at branching sites in the rat thoracic aorta rather than in neighboring non-branching areas, while the Ca^2+^ response to histamine displayed the opposite behavior [144]. Similarly, ATP evoked a robust increase in [Ca^2+^]_i_ in ~90% of mouse aortic endothelial cells, while acetylcholine, substance P and bradykinin were significantly less effective [145]. The onset of the Ca^2+^ response to acetylcholine was constrained by the expression of M3 receptors on a limited fraction of endothelial cells [30]. Interestingly, the endothelial Ca^2+^ response to shear stress in the rat carotid artery was also not uniform and limited to a discrete number of cells [29].

To gain further insight into the heterogeneity of endothelial Ca^2+^ signals, Wilson and McCarron designed a miniature fluorescence endoscope that was developed around a gradient index (GRIN) lens to measure the Ca^2+^ activity in hundreds of endothelial cells in pressurized arteries [28]. Using this approach, they found that, in the rat carotid artery, low concentrations of acetylcholine (≤3 µM) elicited a Ca^2+^ signal in only a minority of endothelial cells. Increasing the concentration of acetylcholine up to the mid-to-high micromolar range increased both the number of activated cells (i.e., more endothelial cells were recruited) and the amplitude of the Ca^2+^ response within each single endothelial cell [28]. The concentration-dependent increase in the amplitude of the Ca^2+^ response of each endothelial cell spanned an order of magnitude of the acetylcholine concentration [28]. However, the Ca^2+^ sensitivity of the entire endothelial monolayer to acetylcholine spanned over three orders of magnitude of agonist concentration, with low-sensitive and high-sensitive endothelial cells clustered in spatially distinct domains of the vascular intima [28]. This arrangement renders the vascular endothelium highly sensitive to weak signals without undergoing any saturation, yet extraordinarily capable of responding even to increases in input strength [4,26]. Consistently, when individual endothelial cells in the clusters displayed spontaneous Ca^2+^ release events, these were not propagated to adjacent cells. Intracellular Ca^2+^ waves propagated along the vascular intima only when the agonist, i.e., acetylcholine, was applied and activated two or more neighboring endothelial cells within the cluster [28]. These findings demonstrated that clustered endothelial cells, with the same agonist sensitivity, serve as a local sensory platform that perceives extracellular cues only within a specific concentration range and thereby informs the more distant sites about agonist stimulation to regulate cardiovascular functions. Conversely, spontaneous Ca^2+^ activity cannot be propagated to the remaining endothelial monolayer and is rejected as background Ca^2+^ noise [4,26]. Thus, the vascular endothelium can be viewed as a complex mosaic of separate detection sites that are tuned to detect different agonists (e.g., acetylcholine and ATP) and different concentrations of the same agonist. A follow-up investigation has revealed that these detection sites convey the information encoded within the Ca^2+^ signal to a pre-determined ensemble of distant endothelial cells via “short-cut” connections that increase the speed and efficiency of signal transmission along the vascular intima [146]. Disruption of the organization of the endothelial network has been shown to impair Ca^2+^-dependent vascular functions, such as NO-dependent vasorelaxation, in a rat model of prediabetic obesity [17].

Subsequently, the McCarron group demonstrated that the endothelial clusters selectively expressing P2Y2 and M3 receptors are able to interact when the lumen of the pressurized rat carotid artery is stimulated with both ATP and acetylcholine [143]. They found that vascular endothelial cells are able to combine and extract information from multiple sources by generating a Ca^2+^ signal that did not derive from the simple summation or average of ATP- and acetylcholine-induced Ca^2+^ signals [143]. Consistently, while acetylcholine triggered intracellular Ca^2+^ oscillations and ATP induced transient Ca^2+^ signals, the combined application of both agonists induced Ca^2+^ signatures with intermediate features [143]. The ability of the vascular endothelium to generate new Ca^2+^ signals in response to different stimuli targeting distinct clusters of endothelial cells indicates that these small endothelial networks can communicate within the framework of a larger endothelial network, thereby increasing the computational capacity of the endothelial monolayer [26]. The computational screening of the incoming signals carried by endothelial cells is amplified by the interaction between the molecular pathways recruited downstream of distinct signaling systems. In this scenario, not only the spatial distribution but also the kinetics of the endothelial Ca^2+^ signal play a crucial role in the engagement of the most appropriate Ca^2+^-dependent decoder, as outlined in the next sections.

## 4. Local Endothelial Ca^2+^ Signals Regulate Mean Arterial Pressure and Local Blood Perfusion

Endothelial Ca^2+^ signals can induce vasorelaxation by recruiting a variety of Ca^2+^-dependent effectors [10,27,147,148], such as (1) endothelial nitric oxide synthase (eNOS) (Figure 1), which catalyzes the conversion of L-arginine to NO by requiring several cofactors, including CaM, tetrahydrobiopterin (BH4), flavin mononucleotide (FMN), flavin adenine dinucleotide (FAD) and iron protoporphyrin IX (Heme Fe); (2) phospholipase A2 (PLA2), which cleaves AA from the phospholipids that are located on the inner leaflet of the PM and ignites the signaling pathway leading to the conversion of AA to prostacyclin (or prostaglandin I2, PGI2) by cyclooxygenase; and (3) SK_Ca_/IK_Ca_ channels (Figure 1), which are activated by an increase in submembrane Ca^2+^ concentration. NO is primarily responsible for the vasorelaxation of large conduit arteries, while SK_Ca_/IK_Ca_ channels are critical for controlling the diameter of smaller resistance-sized arteries and resistance arterioles [5,10]. Nevertheless, NO also contributes to the regulation of vascular resistance, and NO release can be finely tuned by SK_Ca_/IK_Ca_-channel-mediated EDH [6,10,27]. Conversely, NO is central to the regulation of pulmonary resistance arteries, in which SK_Ca_/IK_Ca_ channels only play a minor role [103,149].

### 4.1. The Selective Coupling of SOCE with eNOS: Indirect Evidence for Orai1-Mediated Ca^2+^ Sparklets in Vascular Endothelial Cells in Large Conduit Vessels

NO has been identified as the first endothelium-dependent vasorelaxant mediator in response to both neurohumoral stimuli and shear stress more than 30 years ago [150,151,152,153,154,155]. Endothelial NO is mainly synthesized by eNOS, also known as NOS3, which is located in Ω-shaped invaginations of the PM that are enriched in caveolin-1 and are termed caveolae [27,156]. Caveolin-1 exerts a tonic inhibitory effect on eNOS, thereby preventing NO production. An increase in submembrane Ca^2+^ levels stimulates CaM to bind to and displace caveolin-1 from eNOS, resulting in NO release. NO, in turn, diffuses to adjoining VSMCs, where it activates a soluble guanylate cyclase (sGC)/cyclic GMP (cGMP)/protein kinase G (PKG) signaling pathway to induce vasorelaxation and reduce blood pressure [6,27,147]. A seminal investigation by the Blatter group on CPAE cells has shown that the endothelial SOCE is selectively coupled to eNOS while being seemingly insensitive to InsP_3_-induced ER Ca^2+^ release [66,157]. In agreement with this finding, endothelial SOCs are also primarily located within caveolae, and this structural feature may have profound pathological implications [61,158,159]. Reducing membrane cholesterol can impair caveolar integrity and inhibit agonist-induced SOCE in pulmonary artery endothelial cells in a rat model of chronic hypoxia-induced pulmonary hypertension [160]. That SOCE is the major Ca^2+^ source for eNOS activation in large conduit vessels was also demonstrated by the reduction in NO-dependent vasorelaxation caused by the genetic deletion [161] or knockout [121,122] of STIM1 and the genetic deletion of TRPC4 [123] in several macrovascular beds (see [61]). The McCarron group showed that flow-induced vasodilation in the rat carotid artery is mediated by the autocrine action of endothelium-derived acetylcholine that triggers intracellular Ca^2+^ oscillations and NO release (Table 1) [29]. Although this investigation did not assess the molecular underpinnings of the Ca^2+^ entry pathway, acetylcholine is known to induce endothelial NO release by activating SOCE [162,163].

Measurement of local store-operated Ca^2+^ signals in native endothelial cells is still lacking. A combination of high-speed TIRF microscopy [164] and endothelial-cell-specific GCaMP2 or GCaMP5 mice [165] should enable the recording of Orai1 and TRPC1/TRPC4-mediated unitary Ca^2+^ entry events in en face vascular preparations. It is also still unclear whether Orai1 or TRPC1/TRPC4 are present at the MEPs that decorate resistance arteries and arterioles in the systemic vasculature and lungs [5]. Nevertheless, SOCE plays a major role in supporting eNOS activation in brain microcirculation [131,134], where NO release in response to various neurotransmitters and neuromodulators is severely hampered by Orai1 (in human) or Orai2 (mouse) inhibition [31,35,43,46,78,79,120]. Interestingly, acetylcholine triggered NO production only in the presence of a functional SOCE in mouse [78] and human [31] cerebrovascular endothelial cells, thereby suggesting a tight coupling between Orai1/Orai2 and eNOS.

### 4.2. Ca^2+^ Sparklets and Ca^2+^ Pulsars Recruit SK_Ca_/IK_Ca_ Channels to Induce EDH-Dependent Vasorelaxation

Endothelial Ca^2+^ sparklets and Ca^2+^ pulsars at or near MEPs can relax overlying VSMCs, reduce vascular resistance and thereby increase local blood flow to downstream capillaries by activating juxtaposed SK_Ca_/IK_Ca_ channels [5,6,9,10,27]. SK_Ca_ (K_Ca_2.3) channels present a single-channel conductance of ~10 pS and are encoded by the KCNN3 gene, whereas IK_Ca_ (K_Ca_3.1) channels show a unitary conductance of 20–30 pS and are encoded by the KCNN4 gene. The opening of SK_Ca_/IK_Ca_ channels is regulated by CaM that is bound to their COOH-terminus: an increase in submembrane Ca^2+^ concentration stimulates CaM to gate both channels with a half-maximal activation (EC_50_) at 300–500 nM and a maximal activation at 1 µM [9,27]. Endothelial SK_Ca_/IK_Ca_ channels are mainly concentrated at MEPs, where they can be activated by local Ca^2+^ sparklets and Ca^2+^ pulsars to mediate endothelial hyperpolarization. The negative shift in the membrane potential V_M_ (up to ~−30 mV) is then propagated to the surrounding VSMCs via MEGJs, thereby reducing the open probability of voltage-gated L-type Ca^2+^ channels and inducing VSMC relaxation by a mechanism known as EDH [5,6,9,10,27]. Studies on rodents have shown that SK_Ca_/IK_Ca_ channel currents were detectable in freshly isolated mesenteric artery endothelial cells [101] and in cerebral [166] and pulmonary [103] resistance arteries. Conversely, they are absent in mouse capillary endothelial cells [167], although preliminary evidence suggests that they are present at the human BBB [119].

#### 4.2.1. Ca^2+^ Sparklets Induce EDH-Dependent Vasorelaxation in Systemic Resistance Vessels and in Brain Parenchymal Arterioles

EDH was first found to be activated by TRPV4-mediated Ca^2+^ sparklets (Table 1) [101,102]. The systemic activation of the TRPV4 channels with GSK induced a dose-dependent reduction in the mean arterial pressure (MAP) in mice, rats and dogs [168]. Subsequent work showed that acetylcholine-induced reduction in blood pressure and vasodilation in small mesenteric arteries were impaired in TRPV4-knocked-out mice [169]. Sonkusare et al. reported that acetylcholine induced EDH-dependent vasodilation of resistance arteries by eliciting TRPV4-mediated Ca^2+^ sparklets via the PLCβ-DAG-PKC-AKAP50 signaling pathway at MEPs [101,102]. Low levels of stimulation were primarily associated with IK_Ca_ activity, whereas three to eight individual TRPV4 channels were identified as responsible for the local Ca^2+^ sparklets driving the vasomotor response [101]. Interestingly, a significant increase in MAP was observed in transgenic mice lacking the endothelial AKAP50 or TRPV4 proteins [170]. Hypertension may disrupt TRPV4 channel coupling at MEPs by downregulating AKAP50 and reducing IK_Ca_ activation through a local elevation in peroxynitrite levels, thereby increasing the MAP [170]. Local Ca^2+^ entry through endothelial TRPV4 channels also stimulated SK_Ca_/IK_Ca_ currents and induced vasodilation in mouse mesenteric arteries exposed to low fluid shear stress [53] and rat cremaster arteries experiencing a decrease in intraluminal pressure [55]. Additionally, InsP_3_-dependent TRPV4-mediated endothelial Ca^2+^ sparklets could be activated to generate vasorelaxant signals and attenuate vasoconstriction of mouse mesenteric arteries upon stimulation of VSMC α1-adrenergic receptors [47]. SK_Ca_/IK_Ca_ currents may also be recruited by unitary Ca^2+^ entry signals through TRPV3 [105,171] or TRPA1 [105] and thereby induce vasodilation of rat cerebral parenchymal arterioles in response to, respectively, carvacrol and 4-HNE (Table 1). Finally, NMDAR-mediated Ca^2+^ sparklets induced dilation of mouse pial arteries by engaging SK_Ca_/IK_Ca_ currents (Table 1), while the vasomotor response to NMDAR stimulation was impaired in a mouse model of Alzheimer’s disease, in *5x-FAD* mice [14]. NMDAR-mediated Ca^2+^ sparklets could also recruit the eNOS to promote NO-dependent vasodilation, but this hypothesis remains to be probed [110,120].

#### 4.2.2. InsP_3_-Driven Ca^2+^ Pulsars and Wavelets Induce EDH-Dependent Vasorelaxation in Systemic Resistance Vessels

EDH-dependent vasorelaxation can also be activated by InsP_3_-driven endothelial Ca^2+^ pulsars and wavelets (Table 1). Acetylcholine was found to recruit endothelial IK_Ca_ channels and induce vasorelaxation of mouse small mesenteric arteries by stimulating Ca^2+^ pulsars at MEPs [80]. This Ca^2+^ signaling pathway also contributes to establishing the myo-endothelial feedback [93]. Endothelial Ca^2+^ pulsars can activate SK_Ca_ and/or IK_Ca_ channels and limit VSMC depolarization and vasoconstriction to sympathetic nerve stimulation in rat cremasteric arteries [48], hamster cremaster arterioles [99] and mouse mesenteric arteries [49]. Endothelial Ca^2+^ wavelets at MEPs may also contribute to myo-endothelial feedback and moderate vasoconstriction by recruiting IK_Ca_ channels in hamster retractor muscle feed arteries [50].

#### 4.2.3. Why Do Local Ca^2+^ Signals at MEPs in Systemic Resistance Vessels Fail to Stimulate Robust NO Release?

Acetylcholine-induced vasodilation of systemic mesenteric arteries also presents a modest NO-dependent component [172,173]. The presence of eNOS at MEPs has been reported [174], but NO diffusion to the overlying VSMC layer is prevented here by the alpha chain of hemoglobin (Hbα), which scavenges NO before it can induce vasorelaxation [175,176]. This signaling microdomain is absent in large conduit arteries and helps to understand why NO plays a minor role in the relaxation of smaller, resistance arteries [93,174]. TRPV4 is loosely coupled with eNOS in resistance arteries [103], which causes only a weak reduction in NO production in TRPV4^-/-^ mice. Therefore, it has been proposed that eNOS stimulation in mesenteric resistance arteries is mainly driven by InsP_3_-induced local Ca^2+^ pulsars (Table 1) [80] or regenerative Ca^2+^ waves (Table 2) [101]. However, NO-dependent vasorelaxation in resistance vessels can also be supported by an STIM1-dependent Ca^2+^ source [121,122]. TRPC3 channels, which can be gated by STIM1 and have been detected at MEPs in rat third-order mesenteric arteries, could replenish the ER and thereby maintain acetylcholine-induced Ca^2+^ signaling and NO release (Figure 1) [5,177,178,179]. In line with this evidence, the MAP is also significantly reduced in transgenic mice lacking the eNOS [180,181], supporting the view that NO may play a role in the control of systemic resistance and MAP.

### 4.3. TRPV4-Mediated Ca^2+^ Sparklets Induce NO Release in Pulmonary Resistance Arteries

NO has long been regarded as the predominant vasorelaxant mediator in the pulmonary circulation [182]. Recent investigations have shown that, in contrast to the systemic vasculature, TRPV4 is preferentially coupled with eNOS to induce vasorelaxation in small, resistance-sized pulmonary arteries [149]. The endogenous agonist, ATP, was shown to elicit TRPV4-mediated Ca^2+^ sparklets that were not preferentially located to MEPs but were rather distributed along the endothelial cell membrane. Local Ca^2+^ entry through TRPV4 channels required caveolin-1 and selectively stimulated the eNOS to produce NO, which in turn reduced resting pulmonary arterial pressure (PAP) [103,183,184]. TRPV4-mediated caveolar Ca^2+^ sparklets were impaired in small pulmonary arteries isolated from a mouse model of pulmonary artery hypertension (PAH) and from PAH patients due to the local formation of peroxynitrite, which dismantles the caveolin-1-TRPV4 channel signaling network [184]. A follow-up investigation revealed that ATP is released through endothelial Panx 1 channels and subsequently activates purinergic P2Y2 receptors to stimulate PKCα and induce TRPV4 channel opening [185].

### 4.4. The Complex Regulation of Cerebral Blood Flow (CBF) by Endothelial Ca^2+^ Signaling

Neurovascular coupling, also known as functional hyperemia, is the mechanism whereby an increase in neuronal activity (NA) leads to the vasorelaxation of feeding arterioles, thereby redirecting local CBF to firing neurons [7,134,147]. Recent studies have unveiled that cerebrovascular endothelial cells can sense both neuronal [167] and synaptic [44,186] activity, thereby generating multiple vasorelaxant signals to increase CBF and maintain neuronal metabolism. Capillary endothelial cells, which are positioned in close contact with neurons in the brain, serve as a “sensory web” that generates an electrical signal in response to NA, thereby informing upstream arterioles to increase CBF locally at the site of neuronal firing [167]. Brain capillary endothelial cells express inwardly rectifying K^+^ (K_ir_2.1) channels that are activated by the modest elevation in extracellular K^+^ that occurs during repolarization of each action potential and translate it into a hyperpolarizing signal that propagates through homo-cellular gap junctions to upstream parenchymal arterioles to cause vasorelaxation (Figure 2) [135,167].

A follow-up study by Longden et al. showed that the complex Ca^2+^ signals induced by NA upon the activation of a G_q_PCR at the arteriole–capillary transitional zone (Figure 2) (see Section 2.4.2 were amplified by K_ir_2.1-mediated endothelial hyperpolarization, which enhanced extracellular Ca^2+^ entry through TRPV4 channels (Figure 2). Capillary Ca^2+^ activity led to the local production of NO (Figure 2), which relaxed overlying pericytes to increase the capillary diameter and redirect CBF to the actively signaling capillary bed [131]. It is still unknown why active brain capillary endothelial cells do not transmit their Ca^2+^ activity to adjacent cells via homo-cellular gap junctions [135], as observed throughout the vasculature (see Section 2.4.2). In addition, NA-induced intercellular Ca^2+^ waves in the distal capillary segment can reach the post-arteriole transitional segment via the TRPA1/Panx1/purinergic signaling pathway (Figure 2) [109], albeit at a lower speed as compared to K_ir_2.1-mediated hyperpolarization. Thakore et al. showed that the endothelial Ca^2+^ wave triggered at the site of NA is converted into a fast hyperpolarizing signal at the transitional segment, where endothelial cells express SK_Ca_/IK_Ca_ channels (Figure 2) [109,118]. This fast electrical signal is locally amplified by K_ir_2.1 channels and is electrotonically transmitted via MEGJs to overlying VSMCs to deactivate voltage-gated L-type Ca^2+^ channels and induce vasodilation [109,118].

The hemodynamic Ca^2+^ response to somatosensory stimulation does not fully develop without the additional contribution of endothelial NMDARs, which are mainly located in parenchymal arterioles in the mouse brain and mediate local Ca^2+^ sparklets that recruit eNOS to produce NO and stimulate EDH via SK_Ca_/IK_Ca_ channels (Figure 2) [14,44,120,186].

Recent investigations have shown that endothelial Ca^2+^ signaling and endothelium-dependent vasodilation are disrupted in neurodegenerative disorders, such as Alzheimer’s disease and chronic traumatic encephalopathy [14,187]. Rescuing endothelial ion signaling with synthetic or physiological agonists that stimulate endothelial Ca^2+^ sparklets, e.g., through TRPV3 or TRPA1 channels, may represent an alternative strategy to restore CBF in these and other brain pathologies [134,188].

## 5. Distinct Spatio-Temporal Patterns of Endothelial Ca^2+^ Signals Regulate Angiogenesis

An increase in endothelial [Ca^2+^]_i_ is the primary signal whereby VEGF, the master regulator of angiogenesis, regulates endothelial cell proliferation, motility, adhesion to the substrate, assembly into bidimensional tubular networks and neovessel formation [8]. VEGF stimulates endothelial Ca^2+^ signals by binding to VEGFR-2 (kdr/flk-1), thereby leading to ER Ca^2+^ release through InsP_3_Rs and lysosomal Ca^2+^ mobilization via TPC2, usually followed by SOCE activation [8,130,189,190]. During angiogenesis, VEGF activates a leading tip endothelial cell to become motile and spearhead a new sprout by migrating outward from the parental capillary toward the VEGF source. Trailing cells then begin to proliferate and migrate behind the tip cell as endothelial stalk cells to elongate the sprout and maintain the physical connection to the parental capillary vessel [8,191]. Recent studies have shown that VEGF can induce distinct Ca^2+^ signatures to induce proliferation or migration by recruiting different Ca^2+^-sensitive decoders. Additionally, neovessel formation during wound healing or tumor growth can be supported by de novo blood vessel formation through a process known as vasculogenesis. Endothelial colony-forming cells (ECFCs) are released into the bloodstream by vascular stem cell niches that are distributed along the vascular wall and home to the site of neovessel formation, where they deliver paracrine signals to support angiogenesis or physically engraft within neovessels [71,192]. Like sprouting angiogenesis, ECFC-mediated vasculogenesis is mediated by intracellular Ca^2+^ signals that adopt distinct temporal profiles in proliferating vs. migrating ECFCs [11].

### 5.1. VEGF-Induced Intracellular Ca^2+^ Oscillations Select Endothelial Tip and Stalk Cells

To explore how endothelial Ca^2+^ signaling finely tunes angiogenesis in vivo, Yokota et al. exploited high-speed, light-sheet microscopy in a transgenic zebrafish line expressing GCaMP7a in endothelial cells [38]. They found that VEGF triggered high-frequency Ca^2+^ oscillations in tip cells budding from the dorsal aorta, which were mediated by VEGFR-2. Trailing stalk cells that migrated behind the tip cells also showed VEGFR-2-dependent Ca^2+^ oscillations, although they were not synchronized with those occurring in the leading cells of the sprouts. Interestingly, VEGF-induced Ca^2+^ spikes also occurred in the adjacent endothelial cells before phenotype selection, when they became spatially restricted to the sprouting endothelial cells by Delta-like 4 (Dll4)/Notch signaling [38]. It was not clear whether the Ca^2+^ waves running along the dorsal aorta before tip/stalk cell selection were synchronized by homo-cellular gap junctions or were independently patterned [38]. These findings are consistent with the dynamic model of sprouting angiogenesis: leading tip cells exhibit higher levels of the Dll4 ligand that enhances their migration rate and induces Notch signaling in the follower stalk cells, which are less motile but exhibit a higher proliferation rate [193,194]. However, although Dll4/Notch-mediated lateral inhibition suppresses pro-angiogenic Ca^2+^ oscillations in the endothelial cells that remain within the parental vessel, it does not attenuate the Ca^2+^ spiking activity in stalk cells after they exit from the dorsal aorta [38]. Using a similar approach, Savage et al. showed that VEGF-induced intracellular Ca^2+^ oscillations in endothelial tip cells require the ER transmembrane protein 33 (TMEM33) and regulate phenotype selection by inducing extracellular-signal-regulated kinase (ERK) phosphorylation. The ERK signaling pathway, in turn, was crucial to induce the expression of several tip cell markers, including DLL4 and flt4, to promote filopodia formation and to stimulate endothelial cell migration [129]. Computational modeling confirmed that higher InsP_3_-dependent Ca^2+^ spiking is associated with tip cell phenotype selection and favors Dll4 expression [195]. Future studies will have to evaluate this mechanism in mammalian capillaries, which represent the main source of neovessels after an ischemic event or during tumor growth. It would also be helpful to assess whether the Ca^2+^ oscillations described by Yokota et al. [38] directly engage the Ca^2+^-dependent machinery driving proliferation (primarily in tip cells) and migration (primarily in stalk cells) or whether a distinct Ca^2+^ signaling activity must take place to stimulate these processes (see also Section 5.2). A fascinating hypothesis is that, during sprouting angiogenesis in vivo, VEGF elicits global Ca^2+^ signals to promote phenotype selection [38], whereas it triggers local elevations in [Ca^2+^]_i_ to induce proliferation or migration [124].

### 5.2. VEGF Stimulates Endothelial Cell Proliferation and Migration through Distinct Ca^2+^ Signatures

A large number of in vitro studies, reviewed in detail in [8], have attempted to understand how VEGF-induced intracellular Ca^2+^ signals regulate endothelial cell proliferation and migration after phenotype selection. A recent investigation has revealed that VEGF stimulates endothelial cell proliferation through repetitive oscillations in [Ca^2+^]_i_, whereas biphasic Ca^2+^ signals were better suited to induce proliferation [37]. By studying porcine aortic endothelial cells (PAECs) and HUVECs, Noren et al. found that, when VEGF concentration is raised from the low to high nanomolar range, the percentage of endothelial cells showing repetitive Ca^2+^ oscillations decreases, while the percentage of endothelial cells displaying an initial Ca^2+^ peak followed by a persistent plateau phase increases (Figure 3) [37]. Interestingly, the higher concentrations of VEGF experienced by the leading tip cells induce migration by engaging myosin light chain kinase (MLCK) (Figure 3), whereas the following stalk cells are exposed to lower concentrations of VEGF and undergo proliferation upon the nuclear translocation of the Ca^2+^-sensitive transcription, nuclear factor of activated T-cells 2 (NFAT2) (Figure 3). In accord, only the prolonged increase in [Ca^2+^]_i_ occurring during the plateau phase induced a sustained activation of MLCK [37], which phosphorylates the myosin light chain to promote stress fiber formation and contraction during cell movement [8]. Since the average [Ca^2+^]_i_ achieved in proliferating and migrating endothelial cells was in the same range, the authors concluded that the Ca^2+^ waveform determines the endothelial cell behavior. By simultaneously measuring Ca^2+^ activity and MLCK activation, they showed that a single Ca^2+^ transient is not sufficient to recruit MLCK, whereas a persistent increase in [Ca^2+^]_i_ above a precise threshold (5% of the baseline [Ca^2+^]_i_) stimulated MLCK. This low persistent Ca^2+^ signaling event did not occur in spiking cells because the [Ca^2+^]_i_ returns to the baseline after each transient [37]. Conversely, the four members of the NFAT transcription factor family (NFAT1-NFAT4) are the most appropriate decoders of intracellular Ca^2+^ oscillations [196,197]. In the absence of extracellular stimulation, NFAT is retained in the cytosol by the extensive phosphorylation of its regulatory domain [198]. High-frequency intracellular Ca^2+^ oscillations engage calcineurin to fully dephosphorylate NFAT, thereby inducing its massive translocation into the nucleus [196,197]. In accord, Noren et al. showed that repetitive oscillations in [Ca^2+^]_i_, but not biphasic Ca^2+^ signals of the same average amplitude, supported the nuclear translocation of NFAT2 in vascular endothelial cells [37]. They clearly demonstrated discrete events of NFAT activation in response to each Ca^2+^ spike. Furthermore, by interrogating PAECs with a Bayesian decision model, they revealed that, when the cells are exposed to a gradient of VEGF, they can switch from a migratory to a proliferative phenotype depending on the underlying Ca^2+^ waveform, i.e., biphasic vs. oscillatory [37]. These findings demonstrate that the temporal profile of the Ca^2+^ signal is crucial to control the outcome of VEGF stimulation on endothelial cell behavior. In addition, it reinforces the view that vascular endothelial cells may display multiple Ca^2+^ signatures during sprouting angiogenesis (see Section 5.1). For instance, VEGF could first elicit intracellular Ca^2+^ oscillations that promote phenotype selection (tip vs. stalk cells) and then a novel pattern of repetitive Ca^2+^ spikes to promote proliferation in stalk cells or a different biphasic increase in [Ca^2+^]_i_ to induce migration in the leading tip cells. Interestingly, stromal-derived factor-1α (SDF-1α), a chemokine that is released following a drop in O_2_ tension to stimulate endothelial cell migration, elicits biphasic Ca^2+^ signals, but not intracellular Ca^2+^ oscillations, in several types of vascular endothelial cells [190].

However, a number of issues warrant further investigation. In other cell types, Orai1 is physically coupled with calcineurin and NFAT1 through the scaffolding protein AKAP79 (A-kinase anchoring protein 79) and hence boosts NFAT1 activation by generating local Ca^2+^ nanodomains [199]. Orai1 stimulates the nuclear translocation of NFAT2 in vascular endothelial cells [8,61], such as HUVECs [19,200]. However, it is still unclear whether and how local Ca^2+^ entry events through endothelial Orai1 channels activate NFAT2. In addition, a previous investigation showed that NFAT2 regulates VEGF-induced HUVEC migration and bidimensional tube formation [201]. Additional experiments could be carried out to assess the waveform of the VEGF-induced Ca^2+^ response in the HUVEC cell line employed in [201].

### 5.3. Distinct Ca^2+^ Signatures Control ECFC Proliferation and Migration

The discovery that ECFCs can be mobilized into the circulation either to support vascular regeneration after an ischemic insult [192] or to promote neovessel formation in growing tumors [202] led to the investigation of the role of Ca^2+^ signaling in their angiogenic activity [11]. Dragoni et al. found that VEGF, at low nanomolar concentrations, elicits intracellular Ca^2+^ oscillations, the frequency of which is decreased as VEGF concentration is increased to the high nanomolar range [39], as reported in PAECs and HUVECs [37]. The spiking Ca^2+^ response to VEGF in circulating ECFCs was mediated by rhythmic ER Ca^2+^ release through InsP_3_Rs and supported by lysosomal Ca^2+^ release through TPC1 and by extracellular Ca^2+^ entry through SOCs [39,111]. Notably, the intracellular Ca^2+^ oscillations induced by 10 ng/mL VEGF in circulating ECFCs were in the same range as the frequency threshold required to recruit the Ca^2+^-dependent transcription factor nuclear factor kappa light chain enhancer of activated B cells (NF-κB) in artificially stimulated vascular endothelial cells (2.5 mHz vs. 5 mHz) [203,204]. In accord, 10 ng/mL VEGF stimulated ECFC proliferation and tube formation by inducing the nuclear translocation of the Ca^2+^-sensitive transcription factor, NF-κB [39]. Interestingly, SOCE in circulating ECFCs requires both Orai1 and TRPC1 [205,206], which is functionally coupled to NF-κB activation in the endothelial lineage [61,207,208]. Future work will have to assess whether TRPC1 only provides a molecular scaffold to couple NF-κB to local Ca^2+^ entry through Orai1, or whether it also contributes a pore-forming subunit to the SOCE machinery [61]. Moreover, previous studies have shown that the [Ca^2+^]_i_ threshold that each Ca^2+^ spike must reach during an oscillatory train to induce the nuclear translocation of NF-κB in vascular endothelial cells is ~180 nM [209,210]. Whether this [Ca^2+^]_i_ threshold must also be exceeded in circulating ECFCs remains to be determined. This signaling pathway could be exploited for therapeutic purposes [211]. In accord, the human amniotic fluid stem cell (hAFS) secretome, which promotes coronary neovascularization and partially rescues cardiac function in a mouse model of acute myocardial infarction [212,213,214], induced ECFC tubulogenesis by triggering intracellular Ca^2+^ oscillations that promoted the nuclear translocation of NF-κB [215]. Similarly, optical stimulation with the green light of the photosensitive conjugated polymer, regioregular Poly (3-hexyl-thiophene), rr-P3HT, induced repetitive Ca^2+^ spikes that promoted ECFC proliferation and tube formation by inducing the nuclear translocation of NF-κB [41,216]. This approach does not require viral manipulation of endothelial cells and may provide an alternative strategy to induce therapeutic angiogenesis of ischemic disorders by exploiting the high spatio-temporal resolution of light stimulation [217,218,219].

Recruitment of circulating ECFCs by growing neovessels is favored by chemokines, such as insulin-like growth factor-2 (IGF-2) [220] and SDF-1α [221]. As described above for migrating vascular endothelial cells, both IGF-2 [220] and SDF-1α [222] stimulate ECFC migration in vitro and neovessel formation in vivo through a biphasic increase in [Ca^2+^]_i_. Zuccolo et al. showed that SDF-1α stimulates the G_i_-protein-coupled receptor, CXCR4 to recruit PLCβ2, thereby inducing an initial increase in [Ca^2+^]_i_ driven by ER Ca^2+^ release through InsP_3_Rs, followed by a sustained plateau that was mediated by SOCE [222]. This biphasic Ca^2+^ signal stimulates ECFC homing in vivo by recruiting ERK 1/2 and PI3K/Akt [222], which have long been known to support SDF-1α-induced neovascularization of ischemic tissues [223].

These findings strongly support the view that the temporal profile of the Ca^2+^ signal is crucial to specifically control the different processes that take place during sprouting angiogenesis or vasculogenesis. However, it remains to be investigated whether NFAT may also be activated downstream of VEGF-induced Ca^2+^ oscillations and thereby contribute to triggering the transcriptional program responsible for ECFC proliferation. In addition, the spiking Ca^2+^ activity elicited by VEGF in umbilical cord blood-derived ECFCs (UCB-ECFCs) was remarkably higher as compared to their circulating counterparts [71,224,225]. Of note, the oscillatory Ca^2+^ response to VEGF in UCB-ECFCs was triggered by TRPC3-mediated extracellular Ca^2+^ entry [224], and UCB-ECFCs exhibited a higher proliferative capacity as compared to circulating ECFCs [226]. Future investigations could evaluate whether the higher proliferative potential of UCB-ECFCs depends on the higher frequency of VEGF-induced intracellular Ca^2+^ oscillations or on the selective recruitment of specific Ca^2+^-dependent effectors by TRPC3-mediated Ca^2+^ sparklets [225,227]. Finally, the role played by MLCK in ECFC-mediated neovessel formation has not been assessed. Nevertheless, MLCK was found to stimulate cytoskeletal rearrangement and migration in ECFCs [228] and could therefore be targeted by SDF-1α-induced Ca^2+^ signals.

## 6. Conclusions

The spatio-temporal profile of the underlying Ca^2+^ signals enables endothelial cells (as well as their immature precursors, ECFCs) to fine-tune specific vascular functions, thereby maintaining cardiovascular homeostasis. Elucidating the heterogeneity of the diverse mechanisms whereby local, oscillatory Ca^2+^ signals, such as Ca^2+^ pulsars, Ca^2+^ wavelets and Ca^2+^ sparklets, recruit SK_Ca_/IK_Ca_ and eNOS in different vascular beds will also be essential for designing a more effective therapeutic strategy to target them in disease [134,188,211]. Furthermore, little attention has been paid to the spatio-temporal features of the Ca^2+^ signals that recruit other endothelium-dependent vasorelaxant pathways, including PLA2 and cystathionine ɣ-lyase enzyme, which, respectively, produce PGI2 and hydrogen sulfide [155,229]. It has long been known that TRPV4-mediated Ca^2+^ entry can stimulate PGI2 production [230], but whether PLA2 activation requires local Ca^2+^ sparklets at MEPs or a cell-wide increase in [Ca^2+^]_i_ is still unclear. Similarly, it has long been known that VEGF only elicits biphasic Ca^2+^ signals in vascular endothelial cells, while it is now clear that it may also induce intracellular Ca^2+^ oscillations. Future work will have to scrutinize whether endothelial Ca^2+^ oscillations may also be triggered by other growth factors that are not known to elicit spiking Ca^2+^ activity, such as basic fibroblast growth factor, IGF-2 and angiopoietins [8,40,231]. Many other members of the TRP sub-family of Ca^2+^-permeable channels regulate crucial endothelium-dependent functions, including TRP Melastatin 2 (TRPM2), which can prevent endothelial dysfunction [232] and regulates both angiogenesis and vascular permeability [233,234]; TRPV1, which stimulates angiogenesis [191,218] and regulates NF-κB-dependent gene expression [216]; and finally, the mechano-sensitive TRP Polycystin 1 (TRPP1), which regulates blood pressure [235,236]. These channels are predicted to fine-tune endothelial functions by mediating Ca^2+^ sparklets at specific membrane sites where they are tightly coupled with their downstream Ca^2+^-dependent effectors. However, these discrete Ca^2+^ events are yet to be measured. Similarly, local events of lysosomal Ca^2+^ release through TPCs or TRP mucolipin (TRPML) channels, which could either recruit juxtaposed InsP_3_Rs on the ER or regulate vesicular trafficking and autophagy, also deserve to be visualized in the endothelial lineage [64,237]. The fascinating concept that spatially separated endothelial cell clusters are specialized to perceive distinct inputs and transmit them to more distant sites via pre-defined shortcuts, as well as the emergence of non-linear Ca^2+^ signals that are generated by the summation of multiple inputs, needs to be further explored [4,26,146].

The combination of endothelial-cell-specific knockout mouse models and high-speed imaging (e.g., 2P excitation microscopy and 3D light-sheet fluorescence imaging) will be instrumental in assessing how endothelial Ca^2+^ signals fine-tune vascular function in health and can be targeted in disease [6,238]. We also envision that the development of novel models of human organs, such as organ-on-a-chip, living myocardial slices and pluripotent stem-cell-derived 3D organoids and cell spheroids, will be the indispensable step for the therapeutic translation of the endothelial Ca^2+^ machinery [239,240,241,242,243]. Preliminary evidence showed that, in human peripheral arteries, native vascular endothelial cells present a wide variety of Ca^2+^ signals, ranging from discrete Ca^2+^ events to cell-wide Ca^2+^ waves, that can be severely attenuated by cardiovascular disorders [244].

## Figures and Tables

**Figure 1 ijms-24-16765-f001:**
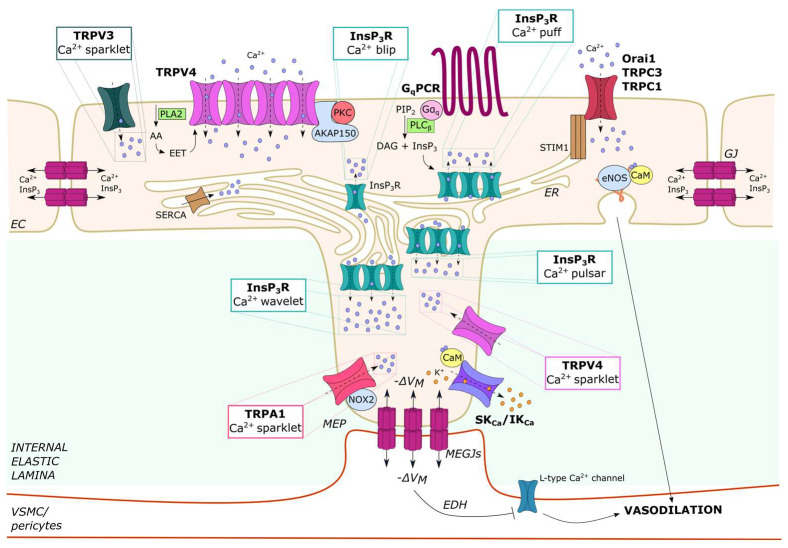
Local Ca^2+^ signals in endothelial cells. Schematic representation of the wall of a microvessel presenting a myo−endothelial projection (MEP) that passes through the internal elastic lamina and establishes physical contact with the overlying vascular smooth muscle cells (VSMCs) in systemic resistance arterioles or pericytes at the arteriole-to-capillary transition zone. Local Ca^2+^ signals in endothelial cells (ECs) are produced by the opening of single InsP_3_R or a cluster of InsP_3_Rs in the ER, which, respectively, generate Ca^2+^ blips and Ca^2+^ puffs. The summation of spatially coupled Ca^2+^ puffs via the mechanism of Ca^2+^-induced Ca^2+^ release leads to global Ca^2+^ waves (not shown). Repetitive openings of smaller clusters of InsP_3_Rs in ER cisternae protruding within MEP originate the Ca^2+^ pulsars. The basal production of InsP_3_ is driven by tonic activation of PLCβ upon the activation of G_q_PCRs located in the PM, such as muscarinic M3 receptors. ER Ca^2+^ refilling during repetitive Ca^2+^ puffs and Ca^2+^ pulsars is likely to be sustained by store-operated Ca^2+^ entry (SOCE), which is mediated by the interaction of the ER-resident Ca^2+^ sensor, STIM1, and one or more Ca^2+^-permeable channels on the PM, such as Orai1, TRPC1 and TRPC3. Local Ca^2+^ signals can be produced also by the activation of Ca^2+^−permeable channels on the PM, known as Ca^2+^ sparklets, including TRPV4, TRPV3 and TRPA1. Ca^2+^ sparklets are mainly coupled with SK_Ca_/IK_Ca_ channels, thereby promoting endothelium-dependent hyperpolarization (EDH), which spreads to overlying VSMCs or pericytes to deactivate voltage-gated L-type Ca^2+^ channels and reduce contractility. The Ca^2+^-sensitive eNOS can also be present in systemic resistance arteries and is widely expressed in brain microcirculation. The Ca^2+^ source responsible for eNOS activation could be provided by the global increase in [Ca^2+^]_i_, although the contribution of Ca^2+^ pulsars has also been postulated. In large conduit arteries and in brain microvascular endothelial cells, eNOS is mainly recruited by SOCE (not shown). Local and global InsP_3_Rs-mediated Ca^2+^ signals could propagate toward adjacent endothelial cells via homo-cellular gap junctions (GJ), thereby generating intercellular Ca_2+_ waves.

**Figure 2 ijms-24-16765-f002:**
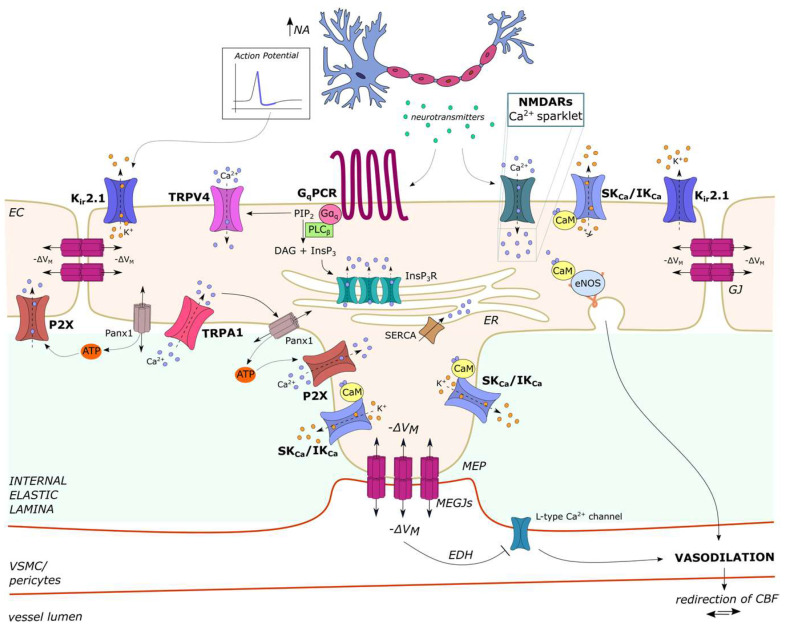
Endothelial Ca^2+^ signaling controls CBF. Synaptic activity can stimulate endothelial G_q_PCRs that induce PIP_2_ hydrolysis, thereby promoting InsP_3_-induced ER Ca^2+^ release and TRPV4-mediated Ca^2+^ entry at capillary level. Here, synaptic activity also results in the local accumulation of extracellular K^+^, which results in K_ir_2.1-mediated endothelial hyperpolarization and amplifies extracellular Ca^2+^ entry. The overall increase in endothelial [Ca^2+^]_i_ stimulates eNOS to produce NO and induce vasorelaxation at the arteriole-to-capillary transition zone. Furthermore, neuronal activity can result in the local accumulation of extracellular ROS that gate brain capillary endothelial TRPA1-channels. TRPA1-mediated Ca^2+^ entry stimulates ATP release via Panx1 channels, thereby triggering an intercellular Ca^2+^ wave that is sustained by the activation of P2X receptors on adjacent endothelial cells. When the intercellular Ca^2+^ wave reaches the arteriole-to-capillary transition zone, it activates SK_Ca_/IK_Ca_ channels and promotes endothelial-hyperpolarization. The negative shift in the V_M_ (−ΔVM; also known as endothelium-dependent hyperpolarization or EDH) spreads through hetero-cellular gap junctions to overlying pericytes, deactivates voltage-gated L-type Ca^2+^ channels and promotes the local increase in CBF. In parenchymal arterioles, synaptically released glutamate can also activate endothelial NMDARs, which engage eNOS and SK_Ca_/IK_Ca_ channels to induce vasorelaxation.

**Figure 3 ijms-24-16765-f003:**
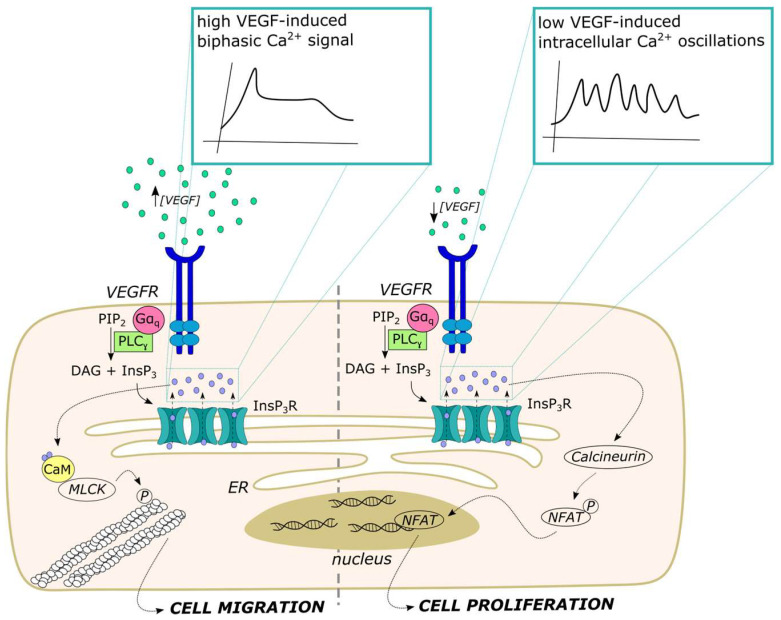
Distinct pro-angiogenic Ca^2+^ signature induced by VEGF in vascular endothelial cells. VEGF elicits an increase in endothelial [Ca^2+^]_i_ by binding to VEGFR-2, a TKR that recruits PLCγ to induce ER Ca^2+^ release through InsP_3_Rs. InsP_3_-evoked ER Ca^2+^ mobilization may be supported by lysosomal Ca^2+^ efflux through TPCs (not shown) and is maintained over time by SOCE activation on the PM (not shown). High doses of VEGF elicit a biphasic increase in [Ca^2+^]_i_ that consists of a rapid Ca^2+^ peak followed by a sustained plateau phase and stimulates endothelial cell migration by activating myosin light chain kinase (MLCK). Low doses of VEGF elicit intracellular Ca^2+^ oscillations that promote endothelial cell proliferation by recruiting calcineurin and inducing the nuclear translocation of NFAT.

**Table 2 ijms-24-16765-t002:** Representative examples of agonist-evoked intracellular Ca^2+^ oscillations in native endothelial cells.

Agonist	Vascular Bed	Function	Mechanism	Reference
Acetylcholine	Rat carotid artery (pressurized)	Flow-dependent vasodilation	InsP_3_-induced ER Ca^2+^ release and an SKF-sensitive Ca^2+^ entry pathway	[29]
Acetylcholine	Mouse carotid artery (en face)	NO-mediated vasodilation	InsP_3_-induced ER Ca^2+^ release, Ca^2+^ entry not assessed	[126]
Acetylcholine	Mouse second-order mesenteric artery (pressurized)	EDH-mediated vasodilation	InsP_3_-induced ER Ca^2+^ release and Ca^2+^ entry	[88]
Acetylcholine	Rat third-order mesenteric artery (pressurized)	Flow-dependent vasodilation	InsP_3_-induced ER Ca^2+^ release and an SKF-sensitive Ca^2+^ entry pathway	[29]
Acetylcholine	Mouse superior epigastric arteries (en face)	Not defined	InsP_3_-induced ER Ca^2+^ release, no Ca^2+^ entry	[90]
Carbachol	Rat tail artery (en face)	Not defined	InsP_3_-induced ER Ca^2+^ release and STIM1-mediated SOCE	[89]
Carbachol	Rat ureteric precapillary arterioles (en face)	NO- and EDH-mediated vasodilation	InsP_3_-induced ER Ca^2+^ release, no Ca^2+^ entry	[83]
ATP	Mouse carotid artery (pressurized)	Not defined	InsP_3_-induced ER Ca^2+^ release and an SKF-sensitive Ca^2+^ entry pathway	[29]
Adenosine	Mouse cremaster muscle arterioles (pressurized)	Not defined	Not defined	[91]
Histamine	Rat lung capillaries (pressurized)	Not defined	InsP_3_-induced ER Ca^2+^ release and Ca^2+^ entry, likely mediated by SOCE	[35,127]
Thrombin	Rat lung capillaries and venules (pressurized)	Lung microvascular permeability	InsP_3_-induced ER Ca^2+^ release, Ca^2+^ entry not assessed	[128]
LPS	Mouse lung microvessels	Lung microvascular permeability	ER Ca^2+^ release through InsP_3_R2 and STIM1/Orai1-mediated SOCE	[20]
VEGF	Zebrafish dorsal aorta and posterior cardinal vein	Angiogenesis	ER Ca^2+^ release through InsP_3_R and STIM1/Orai1-mediated SOCE; plausible contribution of lysosomal Ca^2+^ release via TPC2	[8,38,129,130]

Abbreviations: LPS lipopolysaccharide; SKF: SKF-96365; VEGF: vascular endothelial growth factor.

## Data Availability

Not applicable.

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
