# Peer review of "Cracking the Endothelial Calcium (Ca2+) Code: A Matter of Timing and Spacing"

_ijms, 2023, doi:10.3390/ijms242316765_

Round 1

Reviewer 1 Report

Comments and Suggestions for Authors

The main question that the review is devoted to is "to consider modern ideas about how vascular endothelial cells regulate two important vascular functions - vasomotor movement and angiogenesis, based on the processes of the subcellular spatial profile and/or the temporal pattern of the underlying Ca2+ signal. " 

The topic and the material presented in the work is certainly relevant as it summarizes, in detail and consistently reveals the literature data available at the moment on the role of endothelial Ca2+ signals in the most important vascular reactions.  

The article consistently describes the individual mechanisms, dynamics, spatial distribution, etc. of endothelial Ca2+ signals underlying the performance of their functions by endothelial cells in order to explain how a set of spatially limited (both at the subcellular and cellular levels) Ca2+ signals are used by the intima of vessels to perform complex tasks facing them.  

The article is a completed scientific work that takes into account all modern achievements on the problem being developed. In my opinion, no improvements in the methodology are required in the work. In the future, I would like to wish the authors in their next publication on this topic : a) to consider the mechanisms of formation of endothelial Ca2+ signals disorders in various diseases b) to evaluate the modifying role of environmental factors on endothelial Ca2+ signals and possible mechanisms for correcting these factors.

The conclusion made by the authors on the work is fully consistent with the arguments and evidence presented in the work and does not cause doubts. The conclusion formulated by the authors fully gives an answer to the task set in the work. 

The list of references including 245 sources of literature is sufficient and appropriate.

The three drawings presented in the work as illustrative material are quite visual, made qualitatively, contain all the necessary material in an accessible and easily perceptible form. The drawings made a favorable impression on me.

My general conclusion on the work remains the same: I would like to congratulate the authors on a job well done, after reviewing which I enjoyed!!! The work has a clear structure and is well illustrated. The article includes information from a large number of references, which certainly increases the credibility of the review information presented by the authors. The data presented in the review significantly expand and deepen the scientific information on the subject under study. The work can be published in this edition. 

Author Response

We truly thank the Reviewer for these very nice comments that will push our work further! Thanks, thanks, thanks!

Reviewer 2 Report

Comments and Suggestions for Authors

An excellent and thorough review of the literature around Ca2+ signalling in endothelial cells. The review provides a detailed and critical overview of the different types of ca2+ signaling and how the heterogeneity and fine tuning of these signals, achieved through altered spacio temporal regulation  evokes multiple discrete functions.

The review is extremely long and in my opinion would benefit from reducing the length by removing all elements related to the brain vasculature and focusing on systemic endothelial cells. The brain microvasculature is distinct and the information presented in this review would make a nice additional review article focussed on ca2+ regulation and signalling in brain endothelium

Throughout the article the term ‘naïve’ is used to describe endothelial cells investigated in situ (within the tissue). This term is not appropriate, as the cells are being stimulated so are not naïve. A better term to distinguish these experiments from cultured endothelial cells would be native endothelial cells, or endothelial cells in situ.

An area that is overlooked in the review is prostacyclin. Together with eNOS and EDHF, prostacyclin is an important regulator of vascular tone as well as many other endothelial functions. A few sentences in the relevant sections highlighting what is known about Ca2+ signalling in the production of prostacyclin should be added for completeness.

Minor edits:

Line 84 –change response to responses.

Line 157 – change ‘agonists-evoked’ to ‘agonist-evoked’

Line 158 change ‘an higher‘ to ‘a higher’

Line 162 change ‘agonists-evoked’ to ‘agonist-evoked’

Line 320 ‘Ca2+ sparklets with a spatial of 8 µm2’ a spacial what?

Line 337 In addition, a heteromeric complex comprising STIM1, Orai1, and  TRPC1/TRPC4 and mediating an ICRAC-like current has been described in some endothelial 338 cells [60].

This sentence does not read correctly, perhaps remove ‘and’ after TRPC4?

Line 396 remove ‘s’ from sustains

Line 444 ‘hamster skeletal muscle musculature’ remove ‘muscle’

Line 582 ‘Endothelial Ca2+ sparklets and Ca2+ pulsars at or near MEPs can relax overlying VSMCs, reduce vascular resistance and thereby increase local blood flow to downstream capillaries by recruiting SKCa/IKCa channels [6,7,10,11,26] – please clarify this statement a little. Do the ca2+ sparks activate/open the channels, result in physical movement of the channels to a specific location (MEPs) or both?
